# A Niche for Cowpea in Sub-Tropical Australia?

**Lindsay W. Bell** [1], **Andrew T. James** [2], **Mary Ann Augustin** [3], **Artur Rombenso** [4], **David Blyth** [4], **Cedric Simon** [5], **Thomas J. V. Higgins** [6] and **Jose M. Barrero** [6,*]

1   Commonwealth Scientific and Industrial Research Organisation, Agriculture and Food, Toowoomba, QLD 4350, Australia; Lindsay.Bell@csiro.au
2   Commonwealth Scientific and Industrial Research Organisation, Agriculture and Food, St Lucia, QLD 4067, Australia; Andrew.James@csiro.au
3   Commonwealth Scientific and Industrial Research Organisation, Agriculture and Food, Werribee, VIC 3030, Australia; Maryann.Augustin@csiro.au
4   Commonwealth Scientific and Industrial Research Organisation, Livestock & Aquaculture Program, Bribie Island Research Centre, Woorim, QLD 4507, Australia; Artur.Rombenso@csiro.au (A.R.); David.Blyth@csiro.au (D.B.)
5   Commonwealth Scientific and Industrial Research Organisation, Livestock & Aquaculture Program, Queensland Bioscience Precinct, St. Lucia, QLD 4067, Australia; Cedric.Simon@csiro.au
6   Commonwealth Scientific and Industrial Research Organisation, Agriculture and Food, Black Mountain Science and Innovation Park, Canberra, ACT 2601, Australia; Tj.Higgins@csiro.au
*   Correspondence: jose.barrero@csiro.au

**Abstract:** Pulses have emerged as important rotation crops in Australia. Some are in demand in agricultural production systems due to their high potential market value, because of their roles as grain or forage crops, their nitrogen fixation capability, and because they provide a disease break or improve soil health. While several pulse crops have been identified for winter-season cropping, there are few adapted legumes apart from mungbean that are appropriate for dryland summer cropping. Currently, short-duration crops of mungbean are commonly used, but yields are highly variable and susceptible to drought. Here, we propose that cowpea has the potential to become an alternative rotation crop in dryland summer cropping zones, providing a competitive and profitable alternative pulse crop option where its drought tolerance could enable better performance under inconsistent in-crop rainfall. We demonstrate that cowpea has nutritional properties and putative uses that could prove valuable in emerging plant-based protein and aquaculture markets.

**Keywords:** cowpea; legumes; dryland; rotation crop; food ingredient; aquafeed ingredient





## 1. Introduction

Farming systems in Australia's summer-dominant rainfall zone lack profitable summer-grown grain legume options. Currently, mungbean (*Vigna radiata* (L.) Wilczek) is the main grain legume grown under rainfed conditions throughout southern inland Queensland and northern New South Wales. However, mungbean productivity is highly variable and sensitive to in-crop rainfall, leading to its reputation as a risky crop, and, hence, it is mainly grown in districts with more reliable summer rainfall [1]. Mungbean also provides relatively limited rotational benefits to the farming system, with low biomass and nitrogen fixation, and hosts many of the significant soil-borne diseases in the region. Amongst other grain legumes relatively few options exist. These environments are regularly too dry, and rainfall is too variable for soybean without irrigation. Similarly, other grain legumes (e.g., black gram (*Vigna mungo* (L.) Hepper) or pigeon pea (*Cajanus cajan* (L.) Millsp)) are either limited by market opportunities or suitably adapted genotypes [2]. However, cowpea (*Vigna unguiculata* (L.) Walp) was traditionally grown in many of these regions and is grown in analogous climatic conditions in Africa. Cowpea is considered to be more drought tolerant, provides greater plasticity to variable climatic conditions and provides dual-purpose options as a forage and a grain crop [3]. Nevertheless, for cowpea to become

a viable alternative in the farming system, it would need to have significant agronomic advantages and be able to compete with mungbean on a gross margin basis (yield and price). Cowpea is used, to a limited degree, in these regions but has received relatively little attention, despite its many beneficial attributes. Climate change is predicted to result in a hotter climate and more variable rainfall for sub-tropical Australia [3]. The interest in cowpea, given its higher heat tolerance and better drought avoidance, is partially based on the species being more climate change resilient than other legumes [4]. An ongoing improvement program would be necessary in order to generate and maintain varieties with suitable adaptation.

Furthermore, there is increasing interest in alternative grain legumes for a variety of potential higher value markets. Alternative protein sources for emerging plant-based "meat" products, niche food applications or high-value animal feed formulations (e.g., aquaculture) may present opportunities for alternative grain legumes, such as cowpea. However, cowpea has received relatively little attention as a food or feed ingredient or as starting material for protein isolates for alternative downstream processing and food application. It is anticipated that, rather than traditional commodity whole-bean uses, new high-value uses for cowpea may be developed, such as is the case with the Adzuki type of bean (*Vigna angularis* (W.) Ohwi and Ohashi) for Japanese markets or food ingredients through the use of food processing technologies in order to improve its functional properties. For example, milling and drying of cowpea may be carried out in order to produce a shelf-stable cowpea flour that can be used as a replacement for other flours in a range of food products (e.g., bakery, snack products).

In light of the emerging agronomic and market opportunities, we set out to re-investigate the potential fit of cowpea in Australia. We achieved this by first conducting preliminary ex-ante simulation modelling using APSIM [5] to predict the potential productivity of cowpea compared to other warm-season grain legumes across a diversity of growing environments in sub-tropical and tropical Australia. We also reviewed the nutritive attributes of cowpea through the perspective of its potential for human food processing and aquaculture applications.

## 2. History of Cowpea Research in Australia

Cowpeas were historically cultivated and bred for Australian conditions,, providing both options as a forage and a pulse grain crop. However, the work was reduced in the 1970s due to the susceptibility of cowpea to Phytophthora and the selection of mungbeans as a better pulse option at that time. There have been no reports of Phytophthora outbreaks for over 25 years, due to the existence of resistant varieties but also to changes in rainfall patterns since early 1980s. No new cowpea grain varieties have been developed since "Big Buff" and "Holstein" in the mid 1990s [6,7]. Currently, there is a small production of red cowpea for forage and human consumption. These are grown in marginal cropping environments, while some green-manure and forage varieties are grown in sugarcane rotations. Grain merchants report yields of up to 2 tonnes/ha in cropping of cowpea in Australia. Yields of around 1 tonne/ha are more likely in dryland situations, but additional gains in yield are likely possible with improved varieties and agronomic methods. While few cultivars are currently grown commercially in Australia, cowpea is important internationally, particularly in Africa, and therefore a large cohort of genetic resources exists (see below).

Australia is also home to several native *Vigna* species, which were consumed by the First Nations Australians. The potential value of Australian native legumes of the *Vigna* genus in breeding improved crop varieties for cultivation has been explored and could be important in developing local adaptations [8]. Between 1970 and 1990, efforts were made to make a collection of wild relatives from tropical and subtropical Australia. The aim was to develop a clearer understanding of the natural diversity within the *Vigna* genus, and to evaluate their potential utility in plant genetic improvement of several pulses, such as cowpea and other species of the *Vigna* genus.



## 3. Genetic Resources & Breeding Technologies

The Australian Grains Genebank (part of the Australian Seedbank Partnership; https://www.seedpartnership.org.au/ (accessed on 15 February 2020)) has, in its database, 2470 *Vigna unguiculata* accessions from 71 countries. Approximately 113 accessions were deposited by Australian researchers, including wild races, landraces and breeding lines with diverse origins that were well adapted to different climates and represent both forage and grain types. There is also access to modern cultivars developed for environments, such as in West Africa, that could be of great value under Australian conditions. The International Institute of Tropical Agriculture (IITA) in Nigeria holds the world's largest and most diverse collection of cowpeas, with 15,122 unique samples from 88 countries, representing 70% of African cultivars and nearly half of the global diversity. The phenological plasticity of cowpea varies with the variety [4,9]. Some of the varieties used in small holder agriculture in Africa are highly developmentally plastic, whilst many of the improved varieties from IITA, and the California Blackeye varieties from the USA, are highly determinate and well-suited to mechanical harvesting.

From the point of view of research and breeding, several genetic tools have become available for cowpea that could help fast-track the development of new varieties, including a genome sequence [10], gene expression atlas (http://vugea.noble.org (accessed on 1 June 2021) [11]), and multiple mapping populations (including a MAGIC population [12]) developed for QTL and GWAS studies. These tools have been used successfully in the genetic analysis of multiple traits, for example, in mapping nematode-resistance [13] or for grain size [14]. Mutant collections have also been produced [15,16].

There are several transformation protocols available for cowpea [17], one of which was utilised for the development of a transgenic *Bt* cowpea variety that is resistant to the pod-borer insect *Maruca vitrata* (Fabricius), and that was commercially released in Nigeria in 2019 [18]. The transformation system uses an *Agrobacterium*-based binary vector containing either *nptII* or *bar* genes for selection [19] and converts up to 1% of the cotyledonary node explants into transgenic plants in the greenhouse [17].

## 4. Predicting the Agronomic Fit of Cowpea

### 4.1. Simulation Modelling Approach & Assumptions

The cropping system simulation model APSIM [5] was used to predict the potential yields of cowpea, compared with other grain legumes (mungbean, soybean), over 40 years (1979–2018) and across several environments spanning the grain-producing regions of subtropical Australia. This was done at seven locations that were selected in order to capture the breadth of Australia's subtropical grains cropping zone in northern New South Wales, southern and central Queensland. The sites that were chosen covered a range spanning over 1500 km north-south and varied in aridity (i.e., ratio of rainfall to potential evapotranspiration) at similar latitudes (Table 1). At each site, long-term climate data was sourced from the Bureau of Meteorology SILO database (https://www.data.qld.gov.au/dataset/silo-climate-database, accessed on 15 November 2019) and representative soils for each location were chosen from the APSoil Database (https://www.apsim.info/apsim-model/apsoil/, accessed on 15 November 2019). In most cases, there was limited or no information on the Plant Available Water-holding Capacity (PAWC) for the soils in the different legume crops, hence, this information was predicted based on knowledge about the relative rooting depth and extraction capacity of various other crops. Cowpea has generally been found to extract soil water from a greater depth and to a greater extent than mungbean [20], and hence was assumed to have a higher PAWC (similar to cereal crops). Where data was available or provided for the crops, this was used.

**Table 1.** Details of climate information (mean and co-efficient of variation (standard deviation/mean) of annual rainfall, summer growing season mean daily maximum and minimum temperatures and aridity (rainfall/potential evapotranspiration) and soil type (APSoil number and Plant-available Water-holding Capacity, PAWC) at each location used for long-term simulations of relative potential yields of cowpea and other tropical grain legumes.

| Location | Climate File No. | Lat | Long | Mean Annual Rain (mm) | CV of Annual Rainfall | Aridity of Summer (Nov–Apr) | Mean Daily Temperatures during Summer (Nov–Apr) (°C) | | APSoil No. | PAWC (mm) | | | |
|---|---|---|---|---|---|---|---|---|---|---|---|---|---|
| | | | | | | | Max. | Min. | | Mungbean | Cowpea | Sorghum | Soybean |
| Biloela | 39006 | −24.379 | 150.516 | 662 | 0.26 | 0.39 | 31.8 | 17.7 | 1186 | 175 | 215 | 230 | 189 |
| Emerald | 35264 | −23.569 | 148.176 | 605 | 0.33 | 0.34 | 33.0 | 19.8 | 1175 | 272 | 272 | 272 | 272 |
| Gatton | 40082 | −27.544 | 152.338 | 783 | 0.28 | 0.54 | 29.7 | 17.5 | 148 | 174 | 249 | 249 | 195 |
| Cecil Plains | 41019 | −27.723 | 151.287 | 643 | 0.26 | 0.34 | 30.0 | 15.9 | | 198 | 255 | 255 | 225 |
| Roma | 43091 | −26.548 | 148.771 | 599 | 0.33 | 0.28 | 32.3 | 17.8 | 125 | 130 | 166 | 153 | 149 |
| Gunnedah | 55202 | −30.954 | 150.249 | 606 | 0.26 | 0.29 | 30.2 | 15.1 | 213 | 165 | 219 | 273 | 219 |
| Coonamble | 51161 | −30.978 | 148.380 | 494 | 0.33 | 0.21 | 31.5 | 16.5 | 168 | 134 | 181 | 181 | 181 |

At each location, simulations were designed to predict the growth and grain yields for two cowpea genotypes differing in phenology (short—cv. Banjo, and long—cv. Red Caloona), three soybean cultivars (longer season—cv. Davis, mid-season cv. MG_4, and short-season cv. Williams) and one mungbean cultivar (cv. Emerald). Each of the genotypes was simulated across a range of nine fortnightly sowing dates from 15 October to 15 February between 1978 and 2018 (a period of 40 years). Simulations were designed to reset to a common starting point on 1 May in the autumn of each year, prior to sowing. Soil water was set to the lower limit of sorghum to reflect crops following a summer cereal crop grown in the previous summer. Hence, the available soil water at sowing varied according to rainfall between this time and the sowing date. All legumes were sown at a density of 35 plants per m$^2$ and a row spacing of 0.25 m.

While the crop modules used here were all available in the APSIM version 7.10, several of these models have received limited or little evaluation across a diversity of sites or growing conditions (www.apsim.info, accessed on 15 November 2019). Hence, initial testing and examination of the code was undertaken in order to ensure that they represented their relative phenological development rates, responses to water stress and the allocation of resources between the crops. Initial testing of the released versions of these models in APSIM found significant doubtful differences between the parameters used in the models, while other crop parameters described key crop attributes that are known to differ between species. For this reason, the following changes were made to the released versions of the cowpea model in order to better represent its growth attributes and phenology, relative to mungbean and soybean; otherwise, all other crop parameters were preserved. The main changes implemented were done so to adjust the parameters related to temperature and water stress impacts on growth and leaf area, based on a series of studies that showed cowpea to display a conservative 'drought survival' strategy compared to mungbean, while soybean displayed a 'productivity maintenance' strategy. Details of the parameters altered were:

- Photosynthetic response to temperature was adjusted to reflect the higher temperature tolerance [21] and higher water use-efficiency of cowpea under water stress [20]. This was implemented by increasing by 4 °C the maximum thresholds at which the radiation use efficiency (RUE) is reduced, increasing the maximum temperature for root advance from 32 to 38 °C, and increasing the maximum temperatures that induce leaf senescence from 34 to 35 °C.
- Cowpea have been shown to exhibit much lower leaf senescence and abscission and, hence, to maintain leaf area for longer under moisture-stressed rainfed conditions than mungbean [22]. To capture this difference, the leaf senescence rate during water stress for cowpea was also reduced relative to mungbean (from 0.05 to 0.02), to represent the capacity of cowpea to maintain leaf area through drought periods.

- Root shoot ratio was increased by 30% during all growth stages, and the root growth rate was increased to reflect a greater allocation to roots in cowpea than mungbean [20].
- The phenological development was adjusted to allow for a longer period of grain filling from 280 to 360 degree-days for cv. Banjo and from 362 to 450 degree-days for Red Caloona. This was based on previous field observation of their relative phenology (Andrew James, unpublished data).
- Initial leaf dry matter in the cowpea model was increased from 0.03 to 0.1, and initial leaf area index (LAI) was reduced from 800 to 200 mm$^2$/plant in order to be consistent with mungbean.

### 4.2. Predicted Relative Productivity and Fit of Cowpea

Simulations reflected the expected differences in phenological development of the different genotypes. However, the longevity of all legumes reported here were often ten days shorter than those reported in the field (Figure 1). The cultivars of Emerald Mungbean and Banjo Cowpea had a similar crop duration ($\pm$3–5 days). However, with earlier sowing, the Banjo spent longer in the vegetative phase, reflecting its known greater sensitivity to daylength than cultivated mungbean varieties. The Red Caloona cowpea was simulated to have a duration that was 7–20 days longer than mungbean due to a longer vegetative phase. This difference was increased when grown under cooler conditions, such as later sowing dates or a higher latitude (further south). Finally, the three simulated soybean cultivars represented a range of crop durations differing in maturity by 10–20 days. Soybean, cv. Davis had a longer duration than MG_4, which was longer than Williams. The main difference between these soybean cultivars was the days to the start of flowering, although these differences diminished with later sowing.

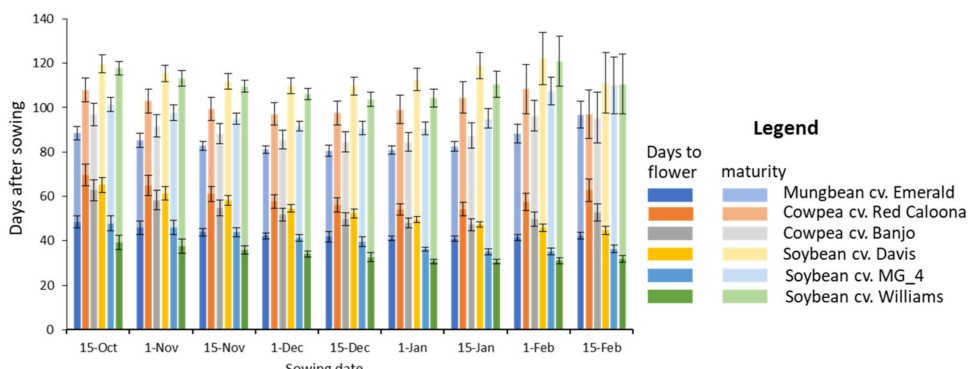

**Figure 1.** APSIM-predicted phenological development (days to flower and maturity) averaged over long-term simulations (1979–2018, $\pm$ standard deviation shown with bars) for mungbean, cowpea and soybean genotypes at one example tropical grain-producing environment (Cecil Plains, Queensland). Similar trends occurred at the other simulated locations.

Across all locations, cowpea genotypes were simulated to achieve similar average grain yields to those in mungbean. These two genotypes typically exceeded the average yields predicted for soybean when grown in these water-limited dryland conditions (Figure 2). In some locations and situations, the average grain yields of cowpea were predicted to exceed those of mungbean. In southern Queensland locations (Roma, Cecil Plains and Gatton), the cowpea genotype Red Caloona was superior to cv. Banjo for earlier sowing windows until mid-late January, after which the shorter duration of Banjo was more favourable. The sowing windows that maximised cowpea grain yields corresponded to those that also maximised average mungbean yields, typically from late December to the start of February.

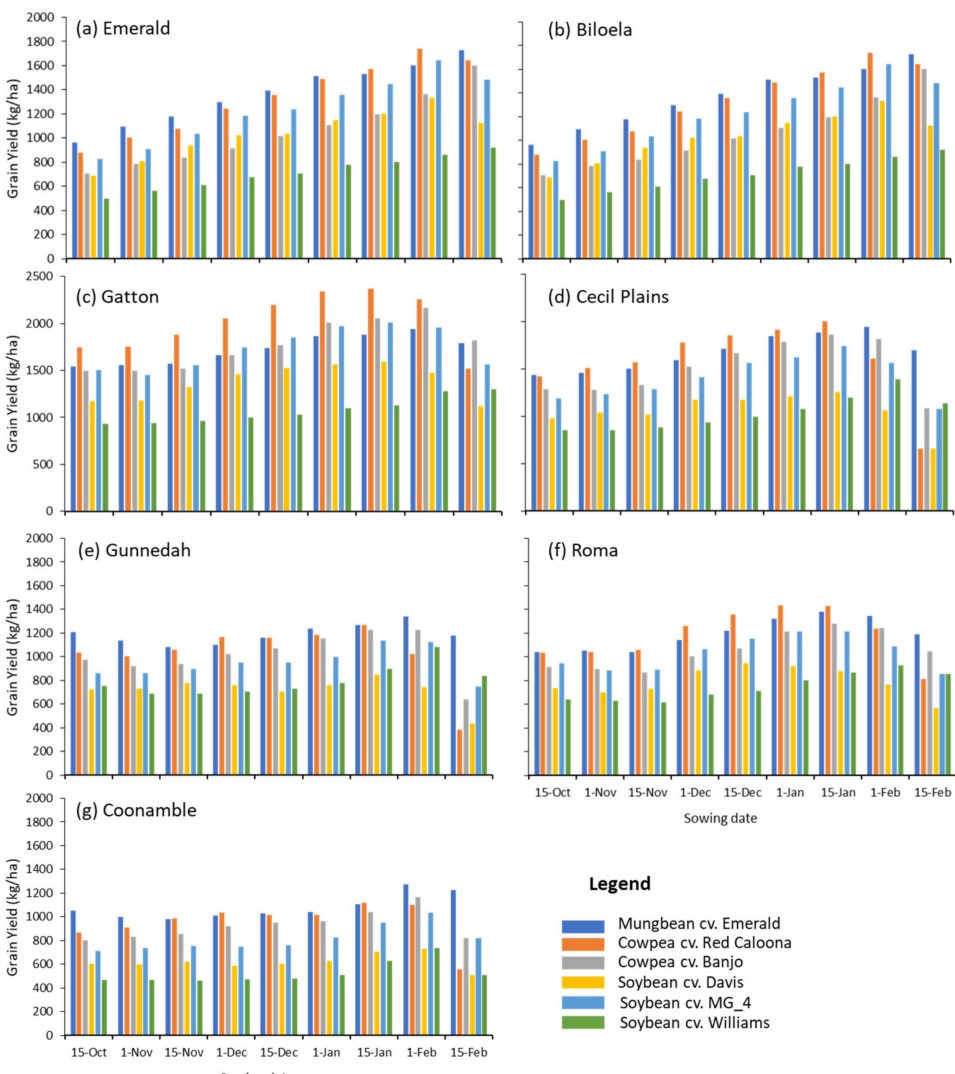

**Figure 2.** APSIM-predicted long-term average (1979–2018) grain yield (kg/ha) for different tropical grain legume genotypes sown at different times at 7 locations spanning the subtropical grain-producing environments of Australia. See Table 1 for details of climate and soils at each site.

While the average yield data simulated here is valuable, the highly variability seasonal conditions in this region is likely to generate a different performance in the different genotypes across wetter vs. drier seasons. For example, it might be expected in wet seasons that soybean, with a higher yield potential, may exceed the yields of the shorter season or more conservative legumes (e.g., cowpea) and vice versa. Hence, to illustrate this, the yield of cowpea and soybean genotypes were compared to those achieved by mungbean in each simulated year. This demonstrates the frequency in which cowpeas would exceed mungbean yields and the size of the predicted yield differences across a wide range of seasonal conditions at each location (see Figure 3). This proved that cowpea could exceed the yield of mungbean by over 500 kg/ha between 30% and 50% of the time, across several situations. The higher yield potentials from cowpeas were most the prospective in southern and central Queensland locations with mid or late season sowing dates. There was less frequent advantage in early season sowings, or later sowing windows in more southerly locations. These planting situations are currently considered risky for mungbean, and it does not seem that cowpeas offer any advantages in these niches at this stage. The analysis also shows that yields of cowpea were less prone to dramatic yield deficits, compared to mungbeans, than the soybeans in drier seasons, as would be expected.

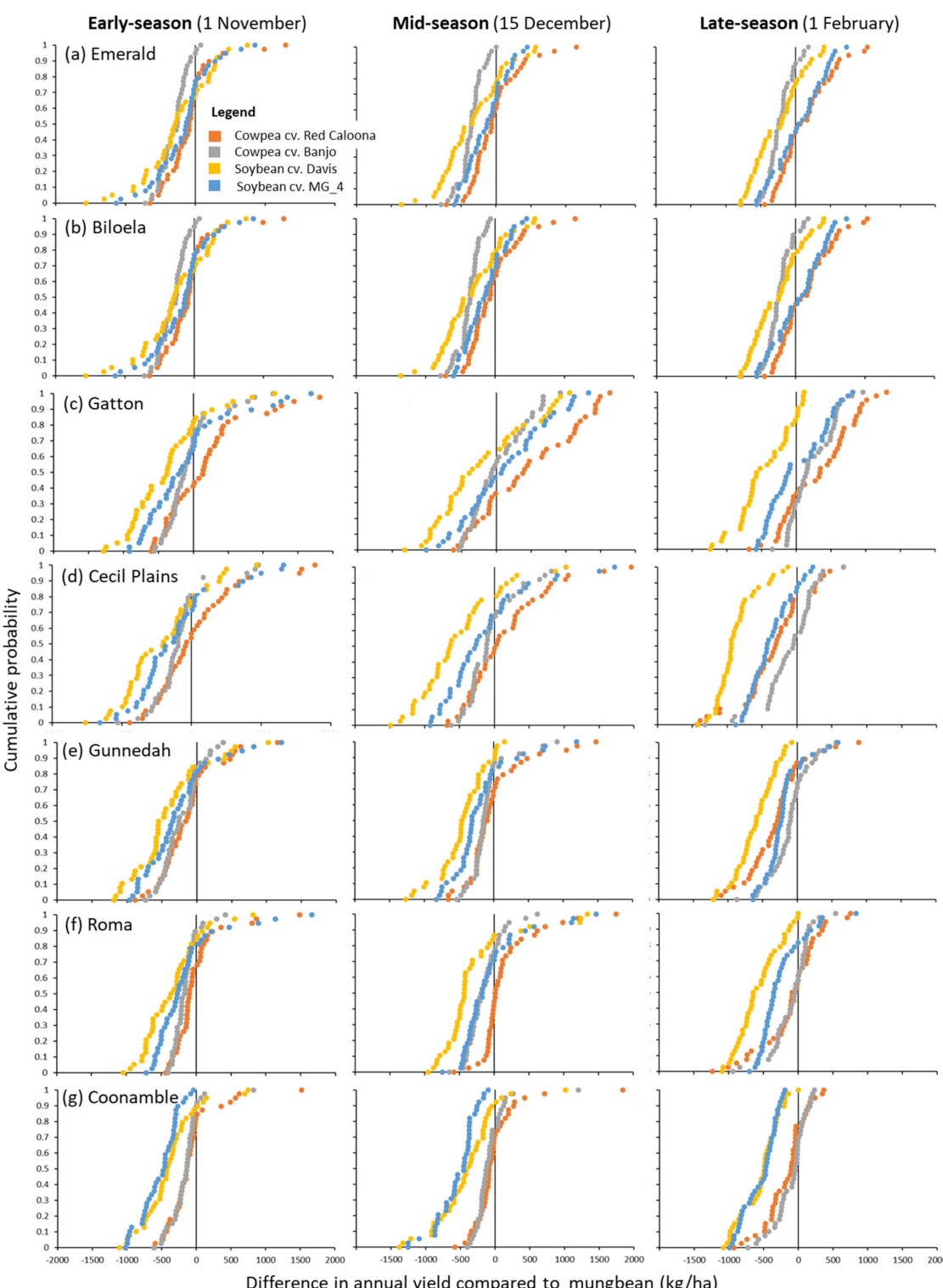

**Figure 3.** Difference in APSIM-predicted annual yields of cowpea and soybeans genotypes compared to those of mungbean (cv. Emerald) over 40 simulated years at different sowing dates (early: 1 Nov, mid: 15 December or late: 1 February) across 7 locations spanning Australia's subtropical grain-producing environments. See Table 1 for details of climate and soils at each site.

*4.3. Further Considerations and Analysis Needs*

While this analysis suggests some positive potential for cowpea, the simulations are using the current APSIM-Cowpea model for which there is limited data to update or recalibrate the model. Hence, there are several uncertainties that need to be considered. Firstly, the phenology, growth and yield of the various genotypes has received little validation across the geographical range considered here. This is particularly relevant in cowpea, where very limited validation or calibration data were used in the model development (see https://www.apsim.info/documentation/model-documentation/crop-module-documentation/cowpea, accessed on 15 November 2019). Hence, more confidence on the accuracy of these models would provide greater certainty to these results. Secondly, most cultivars that have been characterised in the current model and simulated here are old varieties, and it is likely that newer cultivars have made improvements in yield potential and other attributes. This is especially true for cowpea, where the cultivars used here are >30 years old, and it is known that newer genotypes are far superior. However, similar yield gains and phenological differences are known to have occurred in mungbean also. Thirdly, to simulate, with some credibility, the relative physiological and phenological attributes of different legumes and genotypes, assumptions were made about their relative responses to key stresses. While some of these are based on physiological understandings of the relative responses of these crops to water and temperature stresses [23], there are assumptions made (below) that may influence the results both in favour or against cowpea: soil water extraction and root development; plant allocation and grain filling rates; high-temperature tolerance; and water stress tolerance/avoidance and responses.

This analysis also points to some critical knowledge gaps that would to help build our understanding of the fit of cowpea in farming systems. Firstly, characterisation of plant phenology (e.g., crop duration, drivers of development) and resource allocation (e.g., grain filling rates) amongst the diversity of cowpea genotypes. This will help to identify how and when these could potentially fit into different production environments and niches and be used to identify ideal or robust phenology types to target in plant breeding. Secondly, testing or understanding the root growth, soil water extraction and the water stress response differences between cowpea and other legume crops will aid in understanding if cowpea does provide greater resilience to drought and, hence, grain yield or stability advantages in more arid environments. Finally, there is a knowledge gap concerning differences in tolerance to extreme climate conditions (e.g., high temperature) between cowpea relative to mungbean or other species.

## 5. Cowpea Potential for the Food Industry

The nutritional and technological (physical functional) properties of cowpea govern its utility in food. The nutrient composition of cowpea, its flavour and content of desirable bioactive phytonutrients and undesirable anti-nutritive components are some of the important considerations regarding its use in food. Cowpea, like other pulses, is a good source of nutrition [24]. It is a good source of protein, minerals, and phytonutrients (polyphenols) and is low in fat, making it attractive as a nutritional food or ingredient. On a dry basis, the cowpea grain contains 23–32% protein, 50–60% carbohydrate and about 1% fat [25]. In addition to the macronutrients, cowpea contains important minerals, such as calcium (710–1550 mg/kg), zinc (19–49 mg/kg), iron (15–84 mg/kg), magnesium (430–660 mg/kg) [24] and selenium (32–140 ng/g) [26], and micronutrients, such as folate, with reported levels of 1.37–1.86 to $3.67 \pm 0.29$ mg/kg [27,28]. On the other hand, cowpea also contains a number of anti-nutritional components, such as proanthocyanidins (827–3104 mg/kg), phytic acid (2640–15,240 mg/kg) and trypsin inhibitors (3400–67,080 mg/kg), which may affect the bioavailability of the macro- and micro-nutrients [29]. However, some authors have suggested that there was no relationship between proanthocyanidins and mineral or protein digestibility in cowpea [30,31]. As in other crops, the composition of the cowpea varies with the variety and agronomic conditions [32].

A comparison of the proximate composition of cowpea compared to selected pulses is given in Table 2. The protein content of cowpea is similar to other pulses, except that it is lower than that of soybean and lupin (Tables 2 and 3). Cowpea is rich in lysine, but its limiting amino acids are methionine and cysteine. The protein in cowpea comprises 4–12% albumin, 58–80% globulin, 10–15% glutenin and 1–3% prolamin (Table 3). Cowpea is a rich source of carbohydrates, comparable to that of other legumes (Table 3). A significant proportion is comprised of resistant starch and dietary fibre. The fatty acid profile of the fat in cowpea is as follows: 23.9% C16:0, 5.8% C18:0, 12.1% C18:1, 31.1% C18:2, 18.0% C18:3 *n*-3, 2.0% C20:0; 0.9% C20:1; 3.8% C22:0 and 2.1% C24:0 [33]. Cowpea contains many desirable micronutrients, including vitamins (B complex vitamins and Vitamin C) and minerals.

**Table 2.** Proximate composition of selected legumes.

| Pulse | Protein | Fat | Total Carbohydrate | Ash |
|---|---|---|---|---|
| Cowpea (*Vigna unguiculata*) [a] | 22.8 | 1.5 | 61.7 | 3.5 |
| Kidney bean (*Phaseolus vulgaris*) [a] | 22.5 | 1.5 | 61.9 | 3.7 |
| Lentil (*Lens esculenta*) [a] | 24.7 | 1.1 | 60.1 | 3.0 |
| Pigeon pea (*Cajanus cajan*) [a] | 20.4 | 1.4 | 63.7 | 3.7 |
| Groundnut (*Arachis hypogaea*) [a] | 26.3 | 48.4 | 17.6 | 2.3 |
| Soybean (*Glycine max*) [a] | 34.1 | 17.7 | 33.5 | 4.7 |
| Faba bean (*Vicia faba*) [b] | 24.2 | 1.2 | - | 2.7 |
| Mung bean (*Vigna radiata*) [b] | 25.9 | 0.8 | - | 3.6 |

Values represent % on dry weight basis; [a] [34], [b] [35].

**Table 3.** Protein and carbohydrate content and proportion of protein and carbohydrate classes in pulses.

| Pulse | Protein | | | | | Carbohydrate | | | |
|---|---|---|---|---|---|---|---|---|---|
| | Crude Protein (%) | Albumins (% of Protein) | Globulins (% of Protein) | Glutelin (% of Protein) | Prolamin (% of Protein) | Total (%) | Sugars (%) | Starch (%) | Dietary Fibre (%) |
| Cowpea | 24–28 | 4–12 | 58–80 | 10–15 | 1–3 | 42–63 | 3–10 | 35-52 | 11-34 |
| Kidney bean | 17–27 | 15 | 60 | N/a | N/a | 63–74 | 8–11 | 31–43 | 18–30 |
| Chickpea | 19–27 | 8–12 | 53–60 | 18–24 | 3–7 | 52–71 | 3–5 | 30–56 | 6–15 |
| Lentil | 23–31 | 17 | 51 | 11 | 4 | 42–72 | 5–6 | 37–59 | 7–23 |
| Pea | 14–31 | 15–25 | 49–70 | 11 | 5 | 55–72 | 5–12 | 30–49 | 3–20 |
| Lupin | 32–44 | 9–22 | 44–60 | 6–23 | | 47 | 10–16 | 1–9 | 14–55 |

Values given as a range include data from multiple authors [24]. Values are expressed on dry weight basis, percentage of isolated crude protein. N/a—not available.

While cowpea is valued as a source of protein, fibre, starch, vitamins and minerals, the presence of anti-nutritive components (e.g., trypsin inhibitors, protease inhibitors, haemagglutinating activity) constrains cowpea's use in food. Cowpea can also have an undesirable "beany" flavour and the flatus-causing oligosaccharides [29]. Some processing treatments (e.g., boiling, fermentation or heat treatment) can reduce beany flavours and anti-nutritional factors [29].

For the food industry, consistency of raw material composition and functional properties are essential for the reliable production of food ingredients and manufactured food products. Traditional cowpea products include akara (deep-fried fritter), moin-moin (steamed cowpea paste) and cowpea stew, with cowpea also used as a component in weaning foods [36]. Cowpea may be used as an ingredient in many other food applications. Milling of the whole cowpea into flour converts produce into a shelf-stable product which can be used as a replacement for other flours in a range of food products. Conversion of cowpea to powders also reduces post-harvest losses and is a means of managing cowpea that is not sold soon after harvest. Cowpea flours have been demonstrated to be of use in several applications (e.g., bakery, snacks, extruded products) [37]. Alternatively, cowpea

flour may be fractionated to obtain protein-rich or starch-rich fractions. Pulse proteins have foaming, emulsification, water and fat absorption properties, which make them low cost alternatives to conventional protein sources like soy, wheat, and animal protein [38]. An important fractionated product with potential commercial value is cowpea protein isolate. The technological properties (i.e., solubility, foaming, gelling, emulsifying) of the isolate would need to be compared in the application of choice. Processing treatments (e.g., heat treatment, high-pressure processing) may be applied to improve its functionality [39]. It would be useful to examine the potential of protein concentrates and isolates from cowpea as an alternative to presently-used plant-based meat analogues. If suitable, it may provide the impetus for developing a cowpea industry, considering that the global alternative protein and meat substitute market is expected to grow. The emerging market of meat alternatives continues to grow. Technavio forecasts the market to be over $5 billion in 2021 [40].

From a nutritional perspective, cowpea protein isolates showed a net protein retention of 0.7 and digestibility of 87%. The hemagglutinating activity was not detectable and trypsin activity was reduced in the protein isolates. These properties make the isolates an inexpensive source of nutritive protein for the food industry [41]. Additionally, cowpea protein consumption has been found to have favourable effects on proatherogenic serum lipids and apolipoprotein B in human subjects with moderate hypercholesterolemia [42]. Enzymatic hydrolysis of protein isolates has been found to increase antioxidant activity [43]. Recently, Thumbrain et al. (2020) [44] proposed that protein isolates from several cowpea cultivars may have anti-cancer properties.

Cowpea protein has the potential to be a part of the growing market for global protein ingredients and, also, of the alternative protein market if there is research and development to support the production of bland protein concentrates and isolates. The total global protein ingredients market was valued at $36 billion in 2017 and is expected to grow at CAGR 7.2% to reach $51 billion by 2022. This is attributed to increasing demand for both plant-based and animal-based protein ingredients [45]. New sources of plant protein are being sought due to consumer concerns about sustainability and negative consumer perceptions of animal sources of protein. Cowpea has the potential to be a source of protein concentrates, with applications in the meat mimetic and more general food protein market as functional ingredients. The value proposition for value-added cowpea as food ingredients will require a thorough analysis of the agricultural and food value chain across science, industry and business.

## 6. Cowpea in Aquafeeds

Aquaculture nutrition research has focused on protein-rich, and, to a lesser extent, lipid-rich ingredients, whereas those rich in carbohydrate, like cowpeas, are commonly overlooked [46]. This drive towards ingredients rich in protein is associated with the direct relationship between protein and animal growth, the elevated protein demand by aquatic animals, and protein's high contribution to the final feed cost [46,47]. However, in terms of volume, most of the global aquafeed production is used for lower trophic species (e.g., carp and tilapia). These are species that accept higher inclusion levels of carbohydrates than higher trophic species. Although fish and crustacean do not exhibit a demand for carbohydrates (i.e., requirements for specific sugars), this macronutrient plays essential roles in feed formulation and manufacturing. It also has economic relevance on the global aquafeed scale.

Cowpeas are rich in carbohydrates (42–63%) and lacking in crude protein (24–28%) and total lipid (1.5%; Tables 2 and 3). In aquaculture, cowpeas have been investigated in both whole and processed forms, including dehulled, cooked, germinated, extruded, and heat-treated with dry or wet heat [48–56]. As mentioned in the previous section, processing techniques improve cowpea's nutritional value by increasing its digestibility and reducing its antinutritional factors, such as trypsin inhibitors, phytase phosphorus, tannins, phytates, and lectins. For example, autoclaving for 15 and 30 min decreased trypsin inhibitors (TIU)

from 22.8 to 8.4 and 6.9 (TIU/mg), respectively [48]; the same treatment slightly reduced phytase phosphorus and tannins. Heat treatment approaches are practical for reducing heat-labile antinutrients such as lectins. Dry and wet heat treatments at 70 °C and 119 °C reduced lectins from 31 to 8 (µg/g) and to an undetectable level, respectively [49].

The literature on cowpeas in aquafeeds is still incipient, with only nine peer-reviewed publications [48–56] (Summary given in Supplementary Materials Table S1). In black tiger prawns (*Penaeus monodon*), cowpea could replace 16% of the dietary protein requirement and be included at up to 20% in low-cost formulations [50,51]. In whiteleg prawn (*Litopenaeus vannamei*) nutrition, cowpea was identified as a good source of protein and carbohydrate [52]. In finfish, such as tilapia, carp and grouper, cowpea has been included in diets without impairing performance [48,54,55]. Restrictions on the level of cowpea addition are likely due to one or more of the antinutritional-factors such as polyphenols, tannins, trypsin inhibitors, phytates, and lectins, which limit growth in certain fish species.

In the last three decades, aquaculture nutrition has been focused on reducing the reliance on finite wild-caught marine-origin ingredients (i.e., fishmeal and oil), through the inclusion of blends of alternative plant- and terrestrial animal-based ingredients. Fishmeal is still considered to be the benchmark ingredient, serving as a baseline to alternative ingredients (Table 4) [57,58]. Although several publications highlight the suitability of fishmeal-free feeds, the aquafeed industry still prefers fishmeal. However, there is an increasing trend towards the use of feeds without fisheries-based marine ingredients, although most still contain fishmeals that are commonly acquired from rendered aquaculture by-products, such as salmon or tuna meals.

Currently, global aquafeed production is estimated to be about 40–45 metric tonnes (MT), using around 12–16 MT of plant-based ingredients, mostly soybean meal with a value of approximately USD 50 billion [59–61]. In future, this volume is expected to increase, and so too is the demand for alternative ingredients. The selection of an ingredient to for incorporation into a commercial formulation is driven by various factors, including nutritional value, consistency, availability, functionality, and price. All of these attributes of any novel ingredient need to fit the formulation strategies of the aquafeed companies.

Similar to food applications, plant protein concentrates and isolates are used in aquaculture, particularly for high-value carnivorous species (e.g., salmonids, marine fish). Routinely, soybean products enriched in protein, such as soybean protein concentrate and soy protein isolate, are included at a rate of up to 30% inclusion. In Australia, these products are generally imported from the United States or South America and, as such, have a larger environmental footprint than those that could be locally produced [59]. The development of cowpea protein concentrates or isolates could reduce the importation of soybean products for aquaculture. Cowpea protein concentrate was assessed previously in fish (Supplementary Materials Table S1 and Table 4). Lysine and methionine are the major limiting amino acids in aquaculture nutrition [47,57,62,63]. However, the technology chosen and final protein concentration would need to be carefully examined through techno-economic modelling in order to remain cost-effective against commercially available alternatives (Table 4).

**Table 4.** Average protein, lysine and methionine content, and price of fishmeal and selected plant-based ingredients.

| | Protein (%) | Lysine (%) | Methionine (%) | Price of Commodity (USD/MT-2020–2021) [e,f] |
|---|---|---|---|---|
| Fishmeal (herring) [a] | 72 | 7.30 | 2.20 | 1500 |
| Soybean meal [a] | 44 | 2.83 | 0.61 | ~440 |
| Soy protein concentrate [a] | 64 | 3.93 | 0.81 | ~565 |
| Soy protein isolate [a] | 81 | 3.02 | 1.15 | ~1000 |
| Canola meal solvent extracted [a] | 38 | 2.02 | 0.77 | ~400 |
| Canola protein concentrate [a] | 69 | 3.10 | 1.26 | - |
| Cottonseed meal solvent extracted [a] | 42 | 1.60 | 0.58 | 178 |
| Sunflower meal solvent extracted [a] | 32 | 1.20 | 0.82 | ~300–500 |
| Lupin meal [a] | 30 | 1.54 | 0.27 | ~80–210 |
| Wheat flour [a] | 12 | 0.58 | 0.19 | ~240 |
| Barley whole grain [a] | 11 | 0.53 | 0.18 | ~80–120 |
| Dehulled cowpea meal [b] | 21.3–25.6 | 7.0–7.5 | 1.4–2.2 | ~360 |
| Cowpea protein isolate [c,d] | 75 [c] | 6.8 [d] | 1.4 [d] | - |

[a] [47], [b] [31], [c] [64], [d] [41], [e] [65], [f] [66].

## 7. Conclusions and Commercial Opportunities

Here, we explored the potential for cowpea to become an alternative rotation crop in the summer-rainfall environments of sub-tropical Australia. In those regions, agronomists and farmers have identified a need for a new legume, and it is believed that cowpea could provide a resilient medium-duration crop that is able to withstand intermittent drought and inconsistent in-crop rainfall. Because farming practices, mechanisation, and chemistries registration for several food legumes are established in Australia, the expansion of cowpeas should not prove a challenge. The same scenario will be applicable to other similar cropping areas around the world.

Expanding the cowpea crop in Australia, which currently produces 10,000 tonnes with a price of $130 tonne$^{-1}$ on the domestic market, will be a reasonable option. Areas of differentiation for increasing commercial demand for cowpea include cowpea with high folate or iron to outcompete with mungbeans or the development of cowpea varieties with higher protein or oil. Approximately 550,000 ha of dryland sorghum are grown each year in Queensland. If farmers choose to grow dryland cowpea one year out of every four instead of other legume varieties, there will be 140,000 ha of cowpea, which, based on ~1 tonne per ha, means 140,000 tonnes of cowpea will be produced. For aquafeed dietary use in Australia, we predict a market of 25,000 tonnes. Therefore, although the current domestic aquaculture market could only utilize ~20% of production, the overseas market presents an opportunity to generate export revenue. Markets would need to develop, alongside an increase in production. Substantial changes in the market compared with current uses would occur with food processing and the fractionation of the cowpea grain, such as for the burgeoning plant protein market. These are our main conclusions:

- There is abundant germplasm that could be used in a revived cowpea breeding program. This, together with the new genetic resources that are available for cowpea, will speed-up breeding efforts in this crop.
- Our modelling work indicates that cowpea will be a competitive crop in the targeted areas. Across all locations, cowpea genotypes were simulated to achieve similar

average grain yields to those in mungbean, and, in some locations and situations, average grain yields of cowpea were predicted to be larger.

- Regarding uses of cowpea as a food ingredient, we have found that cowpea protein has the potential to be part of the growing market for global protein ingredients and that it can have different functionalities to other food legumes for making differentiated concentrates and isolates.
- We have identified potential application for cowpea and protein enriched products in aquafeed formulations, due to the distinctive composition of the grain. The development of formulation strategies including cowpeas and their protein concentrates could add extra value to this crop.
- Cowpea can offer genetically modified approaches, for example for insect-resistance, which could differentiate cowpea from other similar beans.

**Supplementary Materials:** The following are available online at https://www.mdpi.com/article/10.3390/agronomy11081654/s1, Table S1. Summary of aquaculture nutrition literature evaluating dietary cowpea.

**Author Contributions:** Conceptualization, J.M.B. and T.J.V.H.; Modelling section, L.W.B. and A.T.J.; Food Section, M.A.A.; Aquafeeds section, A.R., D.B. and C.S.; Original Draft Preparation, J.M.B.; Review & Editing, L.W.B., A.T.J., M.A.A., A.R., D.B., C.S., T.J.V.H. and J.M.B. All authors have read and agreed to the published version of the manuscript.

**Funding:** This research was funded by the Commonwealth Scientific and Industrial Research Organisation.

**Acknowledgments:** We thank Meryn Scott for assisting with the literature search related to the use of cowpea in food.

**Conflicts of Interest:** The authors declare no conflict of interest. The funders had no role in the design of the study; in the collection, analyses, or interpretation of data; in the writing of the manuscript, or in the decision to publish the results.

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
