# Peer review of "A Niche for Cowpea in Sub-Tropical Australia?"

_agronomy, doi:10.3390/agronomy11081654_

Round 1

Reviewer 1 Report

The authors did a very good effort to answer my earlier concerns and adapted the MS in that way. I happy to accept the MS in its current revised version.

Reviewer 2 Report

The authors have done a good job of revising the manuscript and they have adequately considered the reviewers comments.  The revised manuscript can now be accepted for publication. 

This manuscript is a resubmission of an earlier submission. The following is a list of the peer review reports and author responses from that submission.

Round 1

Reviewer 1 Report

This paper advocates the use of cowpea in farming systems, including more legumes in the farming systems and their recent use in western foods, including vegan burgers. There have been attempts to introduce cowpea in Australia, and like pigeonpea, they have not met with great success due to susceptibility to Phytophthora. However, the authors believe that new approaches, genetics and demand will address some of the shortcomings that would have limited the adoption of the crop by farmers. There is significant germplasm collection, plus the availability of high yielding cultivars bred overseas, including IITA. They strongly back their argument in favour of cowpea and its fit into cropping systems with modelling work, which is an excellent way of assessing the crop's potential in a new area. Cowpea being a tropical plant, will be grown in the summer season. However, some varieties can also be grown in northern Australia in the dry season, where temperatures are mild. The paper also identifies areas where the current APSIM cowpea model could be weak in simulating cowpea growth and yield, which could be areas for future work. The results discussed, and the work reviewed in the paper will interest the pulse industry, growers, scientists working on the crop in Australia, and similar growing environments of Africa and Asia and hence merit publication.

A few suggestions for improvements are highlighted below.

  1. Figures 1 and 2 will better capture the essence of variability than the bar diagram and comparison with other crops if possible.
  2. There may be a need to consider how climate change could impact the future of the crop even if farmers adopt it.
  3. The crop has highly plastic phenology, a boon in Africa and Asia, but how it will impact Australian growers needs to be discussed. In mungbean under dry conditions, the crop hardly grows and flowers and produces quickly while plant height remains very low for machine harvesting. Will cowpea with even greater phenological plasticity than mungbean will make things worse.
  4. A mention of kind of low and high yields that have been realised from cowpea in Australia in the past will be great in section 2.

Other comments of editorial nature are as given below for the consideration of authors

Abstract

line 17 add coma after ‘used’

Line 19 alternative and not alterative

Introduction

Line 47 high-value and not high value

Line 52 add the before use

Line 57 change ‘This is done …’ with ‘We achieved this…’

History…

Line 71 add comma after environments

Line 91 landraces may be better than land races

Line 99 add comma after varieties

Line 123 ‘Cowpea has’ and not ‘Cowpea is’

                 Could change different to ‘various other’

Lines 127 to 130 change textualize numbers e.g two cowpea, three soybean

Predicting…

Line 158 remove the comma after and

Line 199 Queensland and not Queenland

Line 203 replace useful with valuable

Line 213 remove in after or

Figure 2 legend What is DM/ha? is it oven-dry yield – could include one location in the north.

Line 231 Secondly and not Seconldy

Line 231, add a comma after varieties

Line 232 add commas after cowpea and old

Line 241 High-temperature and not High temperature

Line 247 add a comma after niches

Line 258 replace ‘are’ with ‘is’

Line 272 Add comma after micronutrients

Line276 Do the keywords in the title of the table need to be in capitals?

Line 290 shelf-stable

Line 292 change ‘are’ with ‘is’

Line 301 high-pressure

Line 304 ‘for the development of’ to ‘for developing’

Line 309 add The before ‘Hemagglutinating’

Line 317 add the before ‘potential’

Line 358 add a comma after lectins,

Line 377 high-value

Line 384 change ‘assist with reducing’ to ‘reduces’

Line 123 ‘Cowpea has’ and not ‘Cowpea is’

                 Could change different to ‘various other’

Lines 127 to 130 change textualize numbers e.g two cowpea, three soybean

Predicting…

Line 158 remove comma after and

Line 199 Queensland and not Queenland

Line 203 replace useful with valuable

Line 213 remove in after or

Figure 2 legend What is DM/ha? is it oven dry yield – could include one location in the north.

Line 231 Secondly and not Seconldy

Line 231, add a comma after varieties

Line 232 add commas after cowpea and old

Line 241 High-temperature and not High temperature

Line 247 add a comma after niches

Line 258 replace ‘are’ with ‘is’

Line 272 Add comma after micronutrients

Line276 Do the keywords in the title of the table need to be in capitals?

Line 290 shelf-stable

Line 292 change ‘are’ with ‘is’

Line 301 high-pressure

Line 304 ‘for the development of’ to ‘for developing’

Line 309 add The before ‘Hemagglutinating’

Line 317 add the before ‘potential’

Line 358 add comma after lectins,

Line 377 high-value

Line 384 change ‘assist with reducing’ to ‘reduces’

Reviewer 2 Report

This article, identifying a niche for cowpea production in sub-tropical Australia, uses an interesting holistic approach by combining, climate, agronomic, and nutritional approaches as well diversification for the use of cowpea (e.g., aquaculture). It is in general, well written and to some extent comprehensive. I have some suggestion for additional information that would make the paper stronger. I have put these suggestions in order of appearance in the manuscript and added a ‘*’ when I believe this is a major concern. I suggest the authors carefully consider these comments to develop an improved version.

Line 29-30: is “in-crop” rainfall the correct terminology?

Line 76-86: Is this section relevant?

*Section 4, starting on line 110: Climate data from 40 years period are used. Great, but here I see various missed opportunities. Why are no statistics given for rainfall, temperature (mean, min, max), PAWC,… report median, averages, percentiles,…? Furthermore, this can be used to simulate also variability and hence indicate uncertainty for the data reported in figures 1, 2 and 3.

*Section 4.2, starting at line 177 to 216: this is an important section and following the previous comment this section is also (in contrast to other sections) poorly written (even the English). It is also difficult to follow by reading the text where we can actually see (e.g., tables, figures,…) the data that are discussed. I also miss a good link with the climate data. The quality of the Figs 1-3 can also be better.

Figure 2 and 3: what is the link with the climate data in Table 1 and the “grain producing environments”. This needs to be clarified.

*Section 4.3, starting line 223: Why is the exercise not done by the authors themselves? Why you decided to simulate old varieties (line 231)? This is counterintuitive. Why is not more attention given to identifying, prior to simulations, “robust phenological lines”? This being said, makes lines 250-253 trivial.

*Section 5, starting line 254: why is no information given on important micro-nutrients such as Fe, Zn, Se (and its link with polyphenols and phytic acids in terms of bioavailability for human and animal consumption (see also below))?

Line 369-370 “MT” write as metric tonnes?

Line 381-383: the C footprint data for China appear suddenly and are not really relevant for this paper. You can remove this.

Table 4 can go to suppl. Info.

Table 5: explain the last column.

Line 407: write tonne-1

Line 408: discuss folate and iron higher up in the text.

Line 410: “Ha” should be “ha”.

*Line 413-414: if 25.000 tonnes go to aquaculture, where does the rest of the 140.000 tonnes go? Why is human and animal consumption not discussed in equal detail as the (nice) aquaculture application of cowpea (see also line 429-432).

Line 433-434: This is a conclusion that was, I believe, not discussed in the text. So, you cannot add this to the conclusions.